# Preliminary Study of the Fresh and Hard Properties of UHPC That Is Used to Produce 3D Printed Mortar

**DOI:** 10.3390/ma15082750

**Published:** 2022-04-08

**Authors:** Ester Gimenez-Carbo, Raquel Torres, Hugo Coll, Marta Roig-Flores, Pedro Serna, Lourdes Soriano

**Affiliations:** 1Institute of Concrete Science and Technology (ICITECH), Universitat Politècnica de València, 46022 Valencia, Spain; ratorrem@hotmail.com (R.T.); hucolcar@cst.upv.es (H.C.); pserna@cst.upv.es (P.S.); lousomar@upvnet.upv.es (L.S.); 2Department of Mechanical and Engineering Construction, Universitat Jaume I, 12071 Castellon de la Plana, Spain; roigma@uji.es

**Keywords:** 3D printed concrete, silica fume, setting time, workability, metallic fibers, mechanical properties

## Abstract

Three-dimensional printed concrete (3DPC) is a relatively recent technology that may be very important in changing the traditional construction industry. The principal advantages of its use are more rapid construction, lower production costs, and less residues, among others. The choice of raw materials to obtain adequate behavior is more critical than for traditional concrete. In the present paper a mixture of cement, silica fume, superplasticizer, setting accelerator, filler materials, and aggregates was studied to obtain a 3DPC with high resistance at short curing times. When the optimal mixture was found, metallic fibers were introduced to enhance the mechanical properties. The fresh and hard properties of the concrete were analyzed, measuring the setting time, workability, and flexural and compressive strength. The results obtained demonstrated that the incorporation of fibers (2% in volume) enhanced the flexural and compressive strength by around 163 and 142%, respectively, compared with the mixture without fibers, at 9 h of curing. At 28 days of curing, the improvement was 79.2 and 34.7% for flexural and compressive strength, respectively.

## 1. Introduction

The expanded selection of additives in concrete technology has led to the development of new materials and the possibility of achieving ultra-high-performance fiber-reinforced concrete (UHPFRC). This material is the product of three technologies, self-compacting concrete, fiber-reinforced concrete, and high-strength concrete [1], and was developed with the aim of improving three important aspects, mechanical properties, durability, and workability.

UHPFRC was first developed in France in the 1990s, and, according to the Association Française de Génie Civil (AFGC) [2], this cementitious matrix material has a characteristic 28-day compressive strength of more than 150 MPa, with high flexural strength and ductile behavior. In recent years, there have been small variations in the placement of concrete, with the development of self-compacting concrete and improvements in the techniques for the use of shotcrete, which at the time represented a great advance [3]. Shotcrete can be considered as the ancestor of additive manufacturing. These techniques are the only ones that do not use formwork for the placement of concrete.

In the present work, UHPRC mixtures were developed that could be used in shotcrete as a first step until their dosages could be used to develop additive manufacturing techniques. One of these techniques was three-dimensional concrete printing (3DPC). The challenge presented by the dosages used was that ultra-high-strength concrete is manufactured with large amounts of superplasticizer additive, whose action increases the setting time. However, to use this well-projected material for 3D printing, it was necessary to reduce the setting time, which forced the need to introduce accelerating setting additives into the mixture. This could compromise the mechanical resistance achieved with the mixture [4], which will have to be studied.

3DCP is a material with numerous advantages, but it requires careful dosing. The principal advantages of its use are more rapid construction, lower production costs, and less production of residues, among others [4,5].

Additive manufacturing can be defined as a process that uses technology for automation. With this process, three-dimensional objects can be produced from digital models in a precise way and within a predetermined space. The first research on 3D printing in the construction and architecture industry dates back to 1995, when Pegna suggested incorporating cement-based materials when using these new technologies [6].

Concrete printing can be used to build complex geometric shapes. The components are designed using 3D modelling software [7]. The mixture is printed by controlled extrusion. The concrete needs to have a good degree of extrudability in order to form small concrete filaments. These filaments must bind together to form each layer. One requirement is to be able to build layers without deformation of the successive layers [8].

Lyu et al. [9] noted that the printability of 3DPC includes fluidity, extrudability, buildability, and setting time. The main important factor affecting fluidity in 3DPC is the water content. This concrete used a lower water content, and it required the use of a high-performance water reducing agent. Other factors that can modify the fluidity are the use of mineral admixtures and the effects of aggregate fineness [10].

Extrudability measures the difficulty in the extrusion process. Fresh concrete should be delivered continuously through the nozzle. According to Ma et al. [11], this property is affected by the amount and distribution of the dry mixture.

Buildability measures the degree of deformation and the stability of the printed layers. The material needs to be strong enough to retain its shape and prevent layers from collapsing under its own weight and the gravitational load [12]. Increasing the quantity of aggregates and adding mineral admixtures can improve this property [9].

The setting time for 3DCP requires a compromise between allowing sufficient time to obtain good fluidity and extrudability and sufficient time to obtain early strength.

Zhang et al. [4] noted that the characteristics required for 3DCP often conflict with one another. To obtain a material that is easily pumpable and extrudable, it needs to have low plastic viscosity and optimum yield stress. However, to obtain good buildability, the material needs to have high static yield stress. For all of these reasons, the mixtures should be carefully selected to ensure that they are thixotropic, set suitably, and are densely packed.

Supplementary cementing materials (SCMs) are used to enhance mechanical strength and durability performance, and they also have an influence in the fresh state [13,14,15]. Arunothayan et al. [16] studied the use of fly ash (FA) and ground blast furnace slag (BFS) as substitutes in cement for 3DCP. All of the mixtures contained 30% silica fume (SF) as a constant percentage. The authors demonstrated that the FA facilitated the flow of the mixture, in contrast with the BFS, which reduced the workability. Compressive strength was reduced when SCMs were added; however, the difference with respect to the control was reduced after 90 days of curing.

The present investigation was a preliminary study that explored the use of a mixture of cement, silica fume, superplasticizer, setting accelerator, filler quartz material, and aggregates to obtain 3DPC with high resistance at short curing times. In the first part, different setting accelerators were studied to obtain an adequate setting time, in the second part we worked with the selected accelerator and studied the influence of the setting accelerator percentage on the workability of the fresh mixture and the evolution of flexural and compressive strength. Finally, in the last part of the study, metallic fibers were added to study their influence on resistance. The incorporation of fibers can also improve the bonding and connection between the different layers of placed concrete.

## 2. Materials and Methods

Portland cement type CEM I 52.5R (Lafarge Holcim, Paris, France), which met the specification of the European standards [17], was used in the preparation of mortars. Elkem Microsilica 940 U (Elkem Materials, Pittsburgh, PA, USA) was used as SCM. This material is an undensified silica fume with a density between 200 and 350 kg/m^3^. 

To reduce the cement content and complement the granulometric curve for small sizes, quartz flour (Silbeco, Antwerp, Belgium) was also added; the main characteristics are an SiO_2_ content higher than 98% and density between 200 and 300 kg/m^3^.

X-ray fluorescence analysis was conducted to determine the chemical composition of silica fume and cement, and the results are shown in Table 1.

Figure 1 shows the granulometric curves of cement (CEM), silica fume (SF), and quartz flour (QF).

Two types of siliceous sand were employed, 0.8 fraction with sizes between 0.6 and 1.2 mm and 0.4 fraction with sizes between 0.2 and 0.6 mm. The distribution company was Silicam.

Sika ViscoCrete 225 P (Sika, Baar, Switzerland) was used as the superplasticizing additive. It is a superplasticizer powder water reducer and has a shorter absorption time. The typical dosage varies between 0.05 and 0.5% by weight of binder.

Four types of liquid setting accelerators were used in the first phase of testing to study the behavior of the mixture, in order to finally determine the one that would be used in the second and third phases of the study. Centrament Rapid 500 (MC Company, Scottsdale, AZ, USA) is a chloride-free additive that provides rapid hardening without affecting workability. Sikaset-3 additive (Sika, Baar, Switzerland), according to the manufacturer, is capable of doubling mechanical resistance between 5 and 10 h of curing. Master X-Seed 130 (Master Builders Solution, Mannheim, Germany) is a cement hydration activator agent composed of a suspension of C-S-H nanoparticles. Finally, AKF-63 (IQE, Cardiff, UK) is an aqueous solution of aluminum salts.

The dosage was selected following guidelines from previous studies carried out by our research group [18]. The mortar mixtures are summarized in Table 2. 

All materials were poured into the mixer machine, except for additives (superplasticizer, setting accelerator, and water). The sequence and mixing time of the mixtures are shown in Table 3. The mixing machine employed was an Ibertest model that met the specifications of the European standard [19]. When fibers were incorporated, they were poured into the machine and mixed before the setting accelerator additive.

To determine the setting time, specifically the start of setting of the different mixtures, which occurred in the first phase of the experimental program, two European regulations were taken as reference, UNE-EN 196-3 [20] and UNE-EN 480-2 [21]. Both regulations describe the test procedure, materials, and apparatus to be used to determine setting time. The difference between the two references is the value of the penetration of the Vicat apparatus. In the present study, the start of setting was considered to occur when the needle did not drag the material after extraction and generated a gap in the mortar (Figure 2).

Various investigations have reported that the workability varied depending on the time when the test was carried out [22,23]. In this investigation, the workability was tested at different times after the mixing process was finished, and the European standard was taken for reference [24]. In addition, a smaller cone was used, because it is considered that a smaller sample volume is closer to the amount of concrete that would come out of the nozzle of a 3D printer. The dimensions of the standardized truncated cone-shaped mold, in accordance with European standards [24], were 7 cm inside diameter, 8.5 cm outside diameter, and 4 cm height, with a non-standardized PVC tube 3 cm in diameter and 2.5 cm in height, as shown in Figure 3.

The mechanical strength of 40 mm × 40 mm × 160 mm prismatic mortar specimens was determined according to the normalized standard [19]. Samples were stored in molds in a humid atmosphere until the testing age in the case of short curing times and for 24 h for the rest. At the required age, the specimens were taken from storage and broken by flexure, and each half was tested for compression strength (using an Ibertest machine).

## 3. Results

This section is divided into three subsections; the first part describes the selection of setting accelerator, and the other two parts focus on the chosen additive. 

### 3.1. Selection of Setting Accelerator

The selected percentages of setting accelerator were 1 and 2%, with respect to the weight of cement. Table 4 shows the mortar setting times; the control mortar without setting accelerator had a setting time of more than 90 min. 

The setting accelerator with the shortest initial setting time was AKF-63; this type of additive with aluminum salts was very effective, so it was selected as the setting accelerator for use in successive sections. Kim et al. suggested that aluminate-based activators reacting with cement minerals exhibit fast setting times; they obtained an initial setting time of around 10 min [25]. Other authors, such as Qiu et al. [26], used nano-alumina and modified alcohol amine as raw materials to prepare shotcrete with an initial setting time of only 3 min. The references consulted with regard to the use of C-S-H nanoparticles obtained results very similar to those obtained with Master X-Seed 130. Das et al. [27] reported that the reduction in setting time was a consequence of early strength development of mortar; a reduction of about 43% of the initial setting time was obtained. 

### 3.2. Selection of Percentage of Setting Accelerator

#### 3.2.1. Influence of Percentage of Setting Accelerator on Workability

Before measuring the workability values, we evaluated the initial setting time of the mixture using smaller quantities of setting accelerator; the values obtained were 16 and 13 min for 0.5% and 1.0% and 1.5%, respectively. 

The values of the diameter obtained in the flow table were measured at different times. This time value was a previously defined variable representing the period from the end of the mixing process to the exact moment of raising the cone. After this time, the diameter of the stabilized biscuit was measured in millimeters, as shown in Figure 4.

The evolution of the diameter values with time for the three percentages of setting accelerator and the mixture without setting accelerator is represented in Figure 5.

As can be seen in Figure 3, the diameter of the biscuit obtained depended on the percentage of setting accelerator used. In the mixtures with 1.5% setting accelerator, the diameter was smaller than 10 cm for all times measured, and with 1% setting accelerator, the values were always the same. The setting accelerator promoted a greater degree of hydration, therefore rigidity developed gradually, with a loss of workability [11].

The values obtained in this study were lower than those obtained by other authors as ideal values to obtain good buildability. Tay et al. [28] reported that a slump flow value between 15 and 19 cm in mixtures was optimal for a smooth surface and high buildability.

For future research, the use of other supplementary materials such as fly ash (FA) should be explored, which has demonstrated its ability to increase workability. Liu et al. [29] reported that FA has the property of lubrication and produces a reduction in cement flocculation and, therefore, greater workability.

The study of the mixture in the fresh state, through studies in rheometers, could also be performed to find better dosages for use in 3DCP mixtures. Panda et al. [30] conducted studies on 3DCP using large volumes of AF. They showed that the addition of small amounts of nanoclays improved the behavior of mixtures. The improved performance was associated with the thixotropic property of clay particles, which were responsible for better early age mechanical properties such as yield stress and stiffness.

#### 3.2.2. Influence of Percentage of Setting Accelerator on Flexural and Compressive Strength

The influence of the setting accelerator in the first hours of curing is fundamental to obtaining stability in 3D concrete. In this subsection, the resistance of mortar was measured using 1.5 and 3% setting accelerator. The values of flexural and compressive strength at curing ages of 6, 9, 12, and 24 h and 28 days are listed in Table 5. 

The mechanical strength of 40 mm × 40 mm × 160 mm prismatic mortar specimens was determined according to the normalized standard. The loading speed for the flexural test was 50 ± 10 N/s and the compression rate was 2400 ± 10 N/s.

As can be seen in the resistance results, the setting accelerator had positive effects during the first 24 h of curing. At 28 curing days, both types of resistance were very close to the results of mortar without setting accelerator.

Figure 6 shows the division between the results obtained by mortar with setting accelerator and mortar without additive, i.e., the division between the results of mixtures with 1.5% and 3.0% setting accelerator (R_fi_ or R_ci_) and the mixture containing no additive (R_f0%_ or R_c0%_) for each curing age tested.

The beneficial effects of adding setting accelerator were higher for compressive strength than flexural strength. For both types of resistance, the 3% setting accelerator demonstrated better results. The flexural strength values of mortar with 3% additive at 6 h and 28 days of curing were 3.20 and 1.03, respectively. At 28 curing days, the effect of the additive disappeared.

For compressive strength, the values for mortar with 3% additive at 9 h and 28 days of curing were 6.45 and 1.02, respectively. As with flexural strength, the contribution by the additive to improvements at long curing ages was nil.

The actuation of the setting accelerator in the enhancement of resistance at short curing ages was in agreement with the references consulted [31,32]. 

In some cases, the use of setting accelerator could reduce the strength at long curing times [33], something that did not happen in the present investigation.

Authors such as Min et al. [34] suggested that the presence of setting accelerator promoted the simultaneous hydration of C_3_A and C_3_S at an early age. This process was quicker as the amount of setting accelerator increased. They considered that the reaction effect was effective at an early age, until 12 h.

### 3.3. Incorporation of Metallic Fibers

The bibliography consulted said that the application of 1–3% vol% fibers into 3DCP is applied to obtain sufficient robustness and ductility for structural applications. This reinforcement has been studied with carbon, basalt, glass, or propylene fibers, among others [35,36,37].

In the last part of the present investigation, 2% of metallic fibers by volume was incorporated into the mixture containing 3% setting accelerator. The flexural and compressive strengths of the mixtures with fibers were measured at the same curing ages. Table 6 shows the ratio of mortars with and without metallic fibers (R_i_/R_0_).

The incorporation of metallic fibers enhanced both flexural and compressive strength, and flexural strength improved by a greater extent. The incorporation of fibers in 3D concrete printing is a good solution, because in this system the use of classical steel reinforcement is difficult.

Hambach and Volkmer studied the use of three types of fiber in 3D fiber-reinforced Portland cement paste [38]. They determined that the most effective fiber was carbon fiber at 1 vol%, which achieved flexural strength of 30 MPa. Other fibers (glass and basalt) did not significantly increase flexural strength. Zhu et al. [35] used polyethylene fibers and obtained flexural strength of around 19 MPa.

## 4. Discussion

The results obtained in this study were promising, but this was a preliminary study in which tests were carried out in molds and not in samples obtained by 3D printing. The next step would be to test this dosage using 3D printing equipment and measure the properties of a structure made in this way. Some authors, such as Rehman and Kim [39], pointed out that the compression strength of printed samples could be up to 22% lower than that of samples placed in molds. If the dosage was not suitable for 3D printing, it could be used for the manufacture of shotcrete.

We also believe that rheology studies are essential in future investigations to know how the mixture will behave when used in 3D printing equipment. The behaviour of the fibers in the use of the equipment could also be a critical point; their use could lead to obstruction, segregation, and lack of uniformity in the distribution of them. We consider it essential to expand the study of how to improve the workability of the use of FA without excessively compromising the mechanical resistances. Since FA has a slower pozzolanic reaction than SF and this could mean a decrease in resistances in the early ages of curing.

## 5. Conclusions

The main conclusions of the presented investigation are as follows:

-Mortars with fast setting start were achieved, specifically in a time of 10 min.-The loss of workability over time was pronounced due to the high reactivity of the setting accelerator used.-The percentage of setting accelerator had an influence on the flexural and compressive strength, mainly at short curing ages, but no negative effects at longer curing ages.-With the inclusion of metal fibers, flexural strength of more than 23 MPa and compressive strength of around 76 MPa at 24 h of curing were obtained.

## Figures and Tables

**Figure 1 materials-15-02750-f001:**
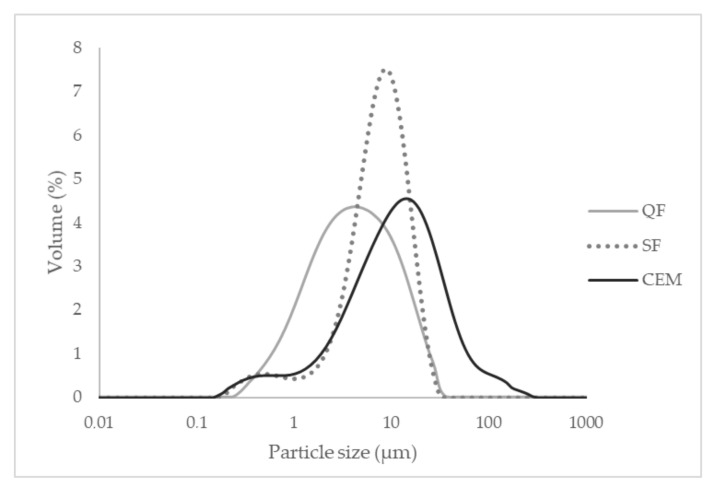
Granulometric curves of cement, silica fume, and quartz flour.

**Figure 2 materials-15-02750-f002:**
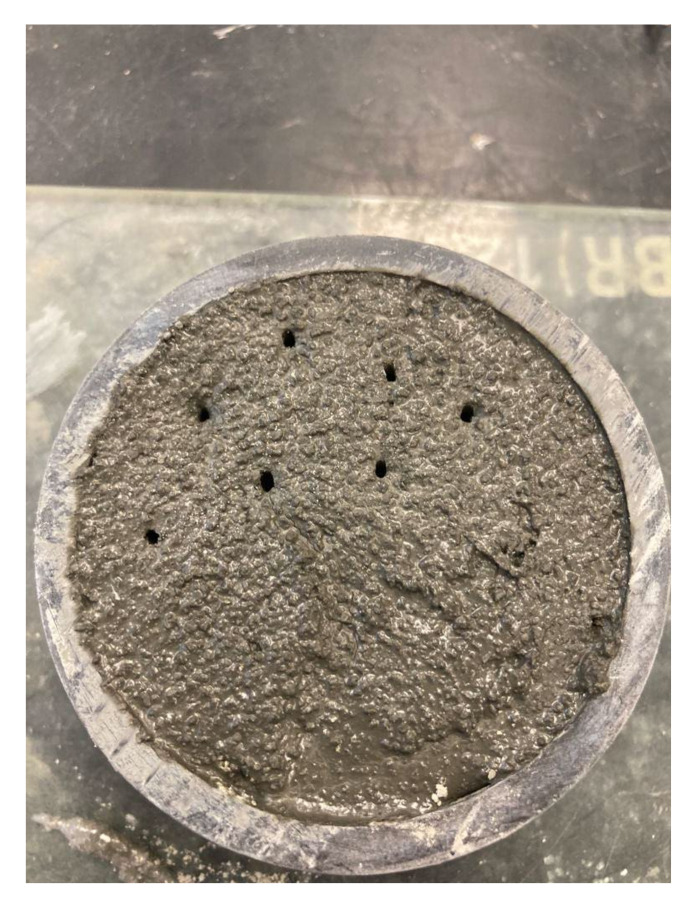
Gap generated in paste indicating the start of setting time.

**Figure 3 materials-15-02750-f003:**
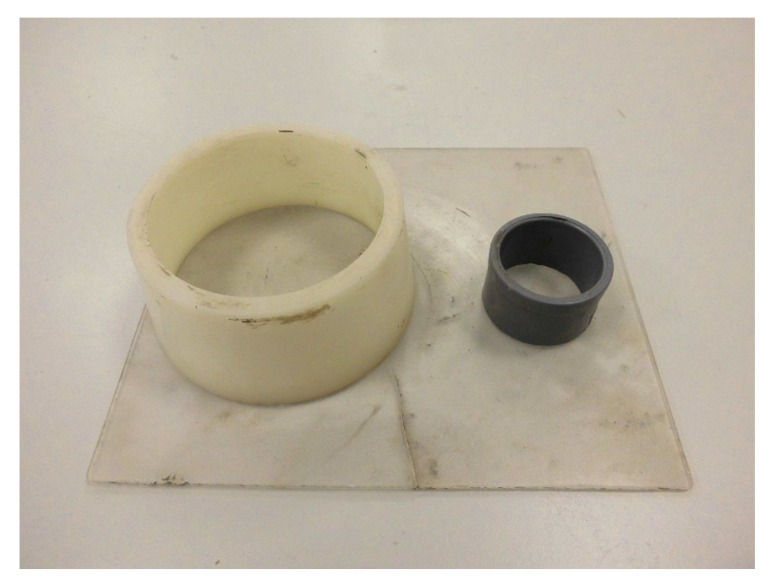
Dimensions of molds used to test workability.

**Figure 4 materials-15-02750-f004:**
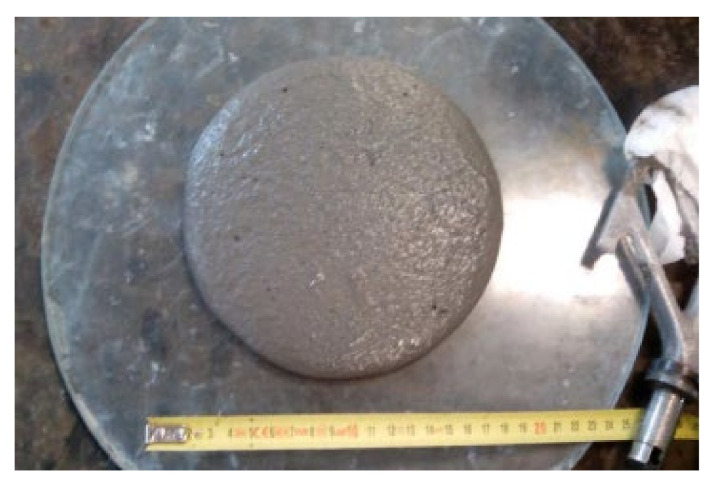
Measurement of the stabilized biscuit.

**Figure 5 materials-15-02750-f005:**
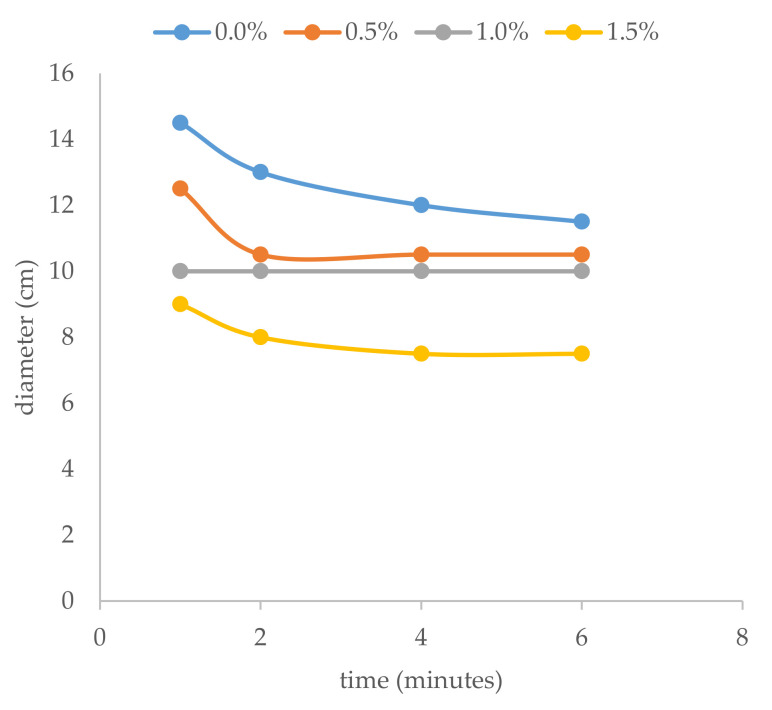
Evolution of workability with time and percentage of setting accelerator.

**Figure 6 materials-15-02750-f006:**
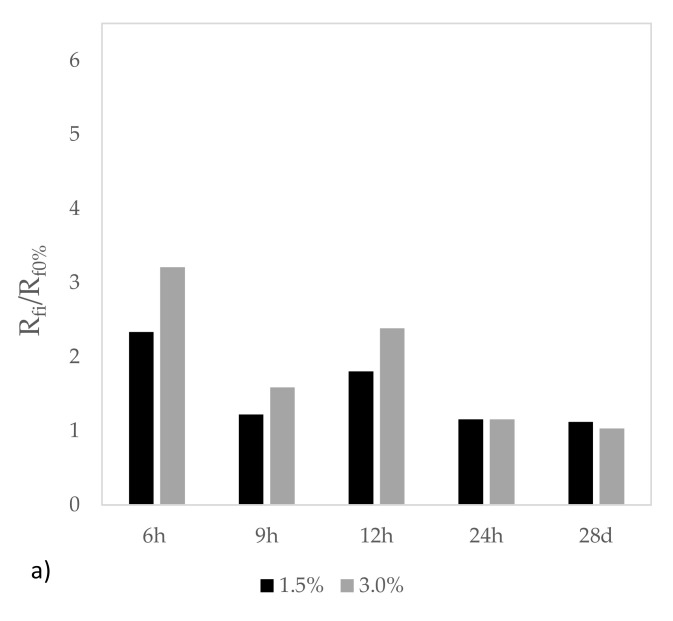
Ratio between mortar with and without setting accelerator: (**a**) flexural strength, (**b**) compressive strength.

**Table 1 materials-15-02750-t001:** Chemical composition of CEM I 52.5R and silica fume (% in weight).

Compound	CEM I 52.5R	Microsilica 940 U
SiO_2_	19.29	95.80
Al_2_O_3_	5.22	0.31
Fe_2_O_3_	3.51	0.14
CaO	61.75	0.38
MgO	2.07	0.10
SO_3_	3.55	0.02
Na_2_O + K_2_O	1.23	0.32
Others	1.42	_
Loss on ignition (950 °C)	1.96	2.93

**Table 2 materials-15-02750-t002:** Mortar dosages.

Material	Dosage (kg/m^3^)
CEM I 52.5R	800
Water	170
Sika ViscoCrete	4
Sand 0.8	562
Sand 0.4	302
Filler addition (quartz flour)	225
Active addition (microsilica)	175

**Table 3 materials-15-02750-t003:** Sequence of material incorporation.

Incorporation in Mixer Machine	Solid Materials (Without Additives)	Water	Additives		Finish Process
			Superplasticizer	Setting Accelerator	
Mixing time	30 s	30 s	150 s	600 s	Variable

**Table 4 materials-15-02750-t004:** Initial setting time of mortar with 1.5 and 2% setting accelerator.

Setting Accelerator	Initial Setting Time (Min)	
AKF-63	1.5%	2%
	10:38	8:00
Sikaset-3	1.5%	2%
	40:02	49:03
Centrament Rapid	1.5%	2%
	60:10	70:30
Master X-Seed 130	1.5%	2%
	65:13	71:00

**Table 5 materials-15-02750-t005:** Flexural strength (R_f_) and compressive strength (R_c_) of mortar.

% Setting Accelerator	Age of Assay	R_f_ (MPa)	R_c_ (MPa)
0.0%	6 h	1.05 ± 0.31	1.92 ± 0.23
	9 h	2.51 ± 0.27	2.20 ± 0.24
	12 h	2.76 ± 0.28	4.70 ± 0.32
	24 h	8.08 ± 0.31	51.52 ± 1.93
	7 d	13.86 ± 0.35	103.31 ± 3.2
	28 d	22.09 ± 0.35	125.77 ± 3.5
1.5%	6 h	2.45 ± 0.29	2.76 ± 0.50
	9 h	3.06 ± 0.27	7.37 ± 0.29
	12 h	4.98 ± 0.51	18.58 ± 0.96
	24 h	9.34 ± 0.78	63.04 ± 0.52
	7 d	14.80 ± 0.67	109.89 ± 1.66
	28 d	24.80 ± 1.02	137.30 ± 1.96
3.0%	6 h	3.37 ± 0.51	7.49 ± 0.97
	9 h	3.98 ± 0.26	14.20 ± 1.2
	12 h	6.58 ± 0.50	28.33 ± 0.79
	24 h	9.34 ± 0.82	60.84 ± 0.94
	7 d	14.25 ± 1.02	105.22 ± 1.21
	28 d	22.28 ± 0.68	129.14 ± 2.10

**Table 6 materials-15-02750-t006:** Ratio of resistance between mortar with and without metallic fibers.

Age of Assay	R_fi_/R_f0_	R_ci_/R_c0_
6 h	2.31	2.71
9 h	2.57	2.20
12 h	2.07	1.22
24 h	2.46	1.25
7 d	2.20	1.26
28 d	1.54	1.35

## Data Availability

Data are contained within the article.

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
