# Peer review of "Preliminary Study of the Fresh and Hard Properties of UHPC That Is Used to Produce 3D Printed Mortar"

_materials, 2022, doi:10.3390/ma15082750_

Round 1

Reviewer 1 Report

The manuscript titled “Preliminary study of fresh and hard properties of UHPC to produce 3D printing mortar” is a very interesting topic. Before its publication on Materials, some minor issues should be clarified by the authors.

  • Title: “Preliminary study of fresh and hardened properties of UHPC to produce 3D-printing mortar”.
  • Line 71: Ma et al. [11]…
  • Figure 1: The terms QF, SF should be defined in the text prior its use.
  • Line 140: summarized in Table 2.
  • Table 2: kg/m3
  • Table 3: what does it means?Can authors clarify it? finish process - variable. 
  • Line 198: Das et al. [27]
  • Line 205: … and 1.5 %, respectively.
  • Table 5: insert the standard deviation.
  • Line 293: … curing. It has no negative effect at longer curing ages.

Author Response

Dear Sir or Madam,

Thank you very much for your useful suggestions on our manuscript; all the errors have been corrected. The responses to all the comments are listed below point by point.

  • Table 3: What does it means? Can authors clarify it? Finish process-variable

As for clarifying why the end time of the process is variable, it is because depending on the type of accelerator additive used together with the superplasticizer it varied. The mixture with the accelerator additive goes from having a dry appearance to a more fluid one as that time varies.

  • Table 5: Insert the standard deviation

As indicated by the reviewer we have added the standard deviation 

Reviewer 2 Report

In this paper, the effects of different kinds of setting accelerators, the content of preferred setting accelerators and the addition of metal fibers on the workability and mechanical properties of 3D printing mortar are studied. 3D printing is a hot research topic in recent years. The article is interesting and clear-cut, but there are some areas worthy of further improvement. Here are some suggestions for the article.

1.The data in Figure 5 shows that the addition of setting accelerator reduces the workability, and the author indicates "The values obtained are lower than the values than other authors expressed as ideal values to obtain a good buildability" in Section 3.2.1, but the author has not put forward relevant measures or methods to improve this situation. It is suggested that the author can quote some literatures here and put forward corresponding solutions.

2.In Section 3.2.2, it can be found from Figure 6 that when the mortar is cured for 28 days, the promoting effect of additives on the strength disappears. Here, I think we can further explain why the influence of additives on mortar weakens or even disappears with the extension of curing. What is the mechanism?

3.Please explain why only 2% volume of metallic fibers was selected as the research object in Section 3.3. Whether 3% and 4% can have better effect, please quote some literatures to improve it.

4.With regard to the description of workability, the following articles also put forward some ideas for reference, which can be cited for a certain prospect.

Effects of wb ratio, fly ash, limestone calcined clay, seawater and sea-sand on workability, mechanical properties, drying shrinkage behavior and micro-structural characteristics of concrete, Construction and Building Materials,2021,311.

Author Response

Dear Sir or Madam,

Thank you very much for your useful suggestions on our manuscript. Your criticisms and comments have been very enriching and have helped greatly to improve it, so we have modified it accordingly. The responses to all the comments are listed below point by point.

  1. The data in Figure 3 shows that the addition of setting accelerator reduces the workability, and the author indicates “The values obtained are lower than the values than other authors expressed as ideal values to obtain a good buildability” in section 3.21, but the author has not put forward relevant measures or methods to improve this situation. It is suggested that the authors can quote some literatures here an put forward corresponding solutions

This comment will be replied to together with the reply to point 4.

  1. In Section 3.2.2, it can be found from Figure 6 that when the mortar is cured for 28 days, the promoting effect of additives on the strength disappears. Here, I think we can further explain why the influence of additives on mortar weakness or even disappears with the extension of curing. What is the mechanism?

We have included the reference [34] and added the following paragraph: “Authors as Min et al [34] suggested that the presence of setting accelerator promoted the simultaneous hydration of C3A and C3S at early age. This process was quicker as the amount of setting accelerator increased. They considered that the reaction effect is effective in early age until 12 hours.

  1. Please explain why only 2% volume of metallic fibers was selected as the research object in section 3.3. Whether 3% and 4% can have better effect, please quote some literatures to improve it

Se utilizó un 2% porque al buscar la bibliografía en los hormigones 3DCP se utilizan porcentajes de fibras entre un 1.0 y un 3%. Two references and an introductory paragraph are added at the beginning of section 3.3: “The bibliography consulted said that the application of 1-3% vol% fibers into 3DCP is applied to obtain sufficient robustness and ductility for structural application. This reinforcement has been studied with carbon, basalt, glass or propylene fibers among others [35-37].

  1. With regard to the description of workability, the following articles also put forward some ideas for reference, which can be cited for a certain prospect. (Construction and Building Materials, 2021,311)

We have included the suggested reference as reference 29 along with another reference (30) and added the following paragraph: “For future research, the use of other supplementary materials such as fly ash (FA) should be explored, which has demonstrated its ability to increase workability. Liu et al. [29] reported that the FA has the property of lubrication and produces a reduction in cement flocculation and therefore greater workability.

The study of the mixture in the fresh state through studies in rheometers can also make to find better dosages and then use them in 3DCP mixtures. Panda et al. [30] have conducted studies on 3DCP using large volumes of FA. They have shown that the addition of small amounts of nanoclays improves the behavior of mixtures. The improved performance is associated with the thixotropic property of clay particles, responsible for better early age mechanical properties such as yield stress and stiffness.”

Reviewer 3 Report

This paper discusses the mix design of 3D printed mortar made of cement, silica fume, superplasticizer, setting accelerator, filler materials, aggregates and fibres. The topic is interesting, but paper need to be revised for more details and can be published after the following revisions:

  • What is the novel aspect of this paper? Has this mix not been used by anyone before?
  • There are many spelling mistakes such as “analayzed” in the abstract. Please check the whole manuscript for spellings/English
  • Use a single acronym UHPFRC consistently in the paper.
  • Figure 6: What is plotted on Y-axis? It is not clearly discussed in the text.
  • Section 3.2.2: Expand more by showing the test-setup, sample sizes, test rate, standards used for testing.
  • The discussion section is very small. Include the discussion in the relevant sections.
  • As the authors have not tested the mix yet for 3D printed, pros and cons should be discussed in the conclusions section for future work.

Author Response

Dear Sir or Madam,

Thank you very much for your useful suggestions on our manuscript. Your criticisms and comments have been very enriching and have helped greatly to improve it, so we have modified it accordingly. The responses to all the comments are listed below point by point.

  • What is the novel aspect of this paper? Has this mix not been used by anyone before?

The novelty is the introduction of the setting accelerator in these mixtures, which is essential for their use as shotcrete or for additive manufacturing. The interaction of various additives in concrete mixtures can change their rheology.

There are many spelling mistakes such as analyzed in the abstract. Please check the whole manuscript for spellings/English

The version that the reviewer read was not yet corrected by the service offered by the publisher, the new version already includes the revisions made by said correction service

  • Use a single acronym, UHPFRC consistently in the paper

The text has been revised to homogenize the nomenclature

  • Figure 6: What is plotted on Y-axis? it is not clearly discussed in the text

The text in the article has been modified and the graph has been improved, the following explanation has been included: “, i.e., the division between the results of mixtures with 1.5% and 3.0% setting accelerator (Rfi or Rci) between the mixture containing no additive (Rf0% or Rc0%) for each curing age tested”.

Round 2

Reviewer 2 Report

My previous comments were carefully addressed.

Reviewer 3 Report

The authors have mostly incorporated the suggested revisions. It can now be accepted.